# Physical Activity, Sport and Physical Education in Northern Ireland School Children: A Cross-Sectional Study

**DOI:** 10.3390/ijerph17186849

**Published:** 2020-09-19

**Authors:** Sinead Connolly, Angela Carlin, Anne Johnston, Catherine Woods, Cormac Powell, Sarahjane Belton, Wesley O’Brien, Jean Saunders, Christina Duff, Orlagh Farmer, Marie Murphy

**Affiliations:** 1Centre for Exercise Medicine, Physical Activity and Health, Sport and Exercise Sciences Research Institute, Ulster University Jordanstown Campus, Shore Road, BT37 0QB Newtownabbey, Ireland; a.carlin1@ulster.ac.uk (A.C.); anneejohnston@outlook.com (A.J.); mh.murphy@ulster.ac.uk (M.M.); 2Department of Physical Education and Sport Sciences, University of Limerick, V94 T9PX Limerick, Ireland; Catherine.Woods@ul.ie; 3Physical Activity for Health Research Cluster, University of Limerick, V94 T9PX Limerick, Ireland; cormacpowell@swimireland.ie; 4Health Research Institute, University of Limerick, V94 T9PX Limerick, Ireland; 5Performance Department, Swim Ireland, Sport HQ, D15 F2CC Dublin, Ireland; 6School of Health and Human Performance, Dublin City University, D15 F2CC Dublin, Ireland; sarahjane.belton@dcu.ie (S.B.); christina.duff4@gmail.com (C.D.); 7School of Education, Sports Studies and Physical Education, University College Cork, T12 KX72 Cork, Ireland; wesley.obrien@ucc.ie (W.O.); 111524727@umail.ucc.ie (O.F.); 8Claddagh Statistical Consultancy Services, Shannon & CSTAR@UL, University of Limerick, V94 T9PX Limerick, Ireland; Jean.Saunders@lineone.net

**Keywords:** youth, school-aged children, adolescents, physical activity, physical education, sports participation, school sport, community sport, enjoyment, social support

## Abstract

Internationally, insufficient physical activity (PA) is a major health concern. Children in Northern Ireland (NI) are recorded as having the lowest levels of PA in the United Kingdom (UK). To date, validated and representative data on the PA levels of NI school children are limited. The aim of this study was to provide surveillance data on self-reported PA, sport and physical education (PE) participation of school children in NI. Differences between genders and factors associated with PA were also examined. A representative sample of primary (*n* = 446) and post-primary (*n* = 1508) children was surveyed in school using validated self-report measures. Findings suggest that PA levels are low, with a minority of children (13%) meeting the PA guidelines (primary pupils 20%, post-primary pupils 11%). NI school children have lower levels of PA, PE and sports participation than UK and European peers. A trend of age-related decline across all the domains of PA was apparent. The data presented highlighted that females are less likely to achieve PA guidelines, children from lower socio-economic background participate in school and community sport less often, and that enjoyment and social support are important variables in PA adherence. Policy solutions that would support implementation e.g., mandatory minimum PE time, whole school approaches to PA promotion and targeted investment in schools, particularly in areas of deprivation and for females, are suggested.

## 1. Introduction

Recent international findings from the Health Behaviour in School-aged Children (HBSC) study show that fewer than one-fifth of adolescents achieve the recommended levels of moderate-to-vigorous physical activity (MVPA) daily [1]. Insufficient physical activity (PA) is a major health problem globally [2]. Given the potential tracking of PA behaviours from childhood into adulthood [3], promoting positive PA behaviours during childhood and adolescence may result in more active adult populations. In addition to providing a foundation for future PA, children’s activity levels and behaviours are also vital to their health and wellbeing in the short term [4].

Consistent with trends observed worldwide [1,5], the majority of young people in Northern Ireland (NI) are not meeting the 60 min of daily MVPA recommended by the World Health Organisation (WHO) and the United Kingdom (UK) Chief Medical Officer [6,7]. Fewer than half of children and adolescents across the island of Ireland are meeting the guidelines [8], and children living in NI are least likely to meet the recommended levels of PA, when compared with other countries across the UK [9].

Declines in MVPA and increases in sedentary time have been observed across childhood (aged 6–11 years) [10,11,12], underlining the importance of targeted interventions in this population to help prevent the observed age-related decline, as children move towards adolescence. Adolescence has been identified as a crucial period when young people recognise the benefits of a healthy lifestyle [13,14]. Furthermore, the experiences of young people during this period are important in terms of nurturing their confidence, competence and motivation for lifelong PA [15,16]. Sport and physical education (PE) provide children with opportunities to be active, both are significant contributors to the development of children and young peoples’ movement competencies, and encouraging the development of active lifestyles [17]. In addition to the PA guidelines, schools in NI are recommended to provide 120 min of PE per week, however to date research suggests that schools are falling short on achieving this, particularly in the primary years [18,19]. Identifying behavioural and environmental barriers and facilitators of PA in this age group is important to understand engagement with PA [20], and may be an appropriate starting point for the design of interventions to increase PA in NI children. Young people from NI consistently report that they value fitness, friendship and fun as being the most important part of their enjoyment and benefit from sport and PA participation [21,22]. A positive association between social support from parents [23,24] and friends [25,26] and PA levels has been reported. Higher levels of enjoyment have been previously associated with increased levels of PA in Irish youth [27,28]. This study sought to explore the associations between enjoyment and social support and NI children’s PA levels.

PA participation encompasses a range of domains, including active travel, PE and sports participation, and the overall decline in youth PA is proposed to be a consequence of declines across each of these domains [10]. The Global Action plan on PA suggests opportunities for PA should be integrated across multiple setting [5]. In order to reverse the trends in PA and sedentary behaviour, countries need to prioritise policies and interventions that target young people, especially females [5,25], in a coordinated way. Unlike the Republic of Ireland (ROI), England, Scotland and Wales, NI does not have a bespoke national PA strategy. Currently, policies relating to the multiple PA domains are sponsored by different NI government departments and agencies. A lack of consistent surveillance data across the full range of domains in NI has hampered conclusions on the effectiveness of policies and initiatives to date aimed at increasing PA levels amongst young people [29]. This study provides baseline data upon which to build future surveillance and to guide policy development.

The Children’s Sport Participation and Physical Activity study 2018 (CSPPA 2018) [29] was extended across the whole island of Ireland and reported comparisons between NI and ROI for the first time [29]. The novelty of this study is that it aims to present findings from the CSPPA 2018 NI cohort only, and report on young people’s participation in PA, PE, school and community sport participation, specifically within the NI context. The study expands on the previously reported descriptive data, providing further statistical analysis and examining additional correlates of participation, namely enjoyment and social support that have not previously been reported.

## 2. Materials and Methods

### 2.1. Study Design

The Children’s Sport Participation and Physical Activity study 2018 [29] was a follow up to CSPPA 2010 [30], to investigate PA, sports participation and PE amongst children and adolescents (aged 10–18 years) across the island of Ireland. In total, 6651 children aged 10–18 took part in the CSPPA 2018 study from across the island of Ireland, with 1954 primary and post-primary students making up the NI sample. The methods and results within this paper focus on a cross-sectional analysis of data from the NI sample.

### 2.2. Sampling

In order to achieve a representative sample, all mainstream primary and post-primary schools from NI were included in the sampling frame. Schools were stratified by four criteria: school gender (male, female or mixed), school location (urban or rural, categorised by population density), and size (small, medium or large, based on total number of pupils) and socio-economic status (high, medium, low). As is standard practice within the NI education system, the socio-economic status of each school was determined by using the percentage free school meals (%FSM) entitlement. FSM is a government initiative for low-income families or those on income support, the lower the % FSM the higher the socio-economic status. Schools that did not meet the criteria relating to age cohort or targeted population were excluded. A total of 51 schools were contacted (20 primary and 31 post-primary) to ensure an equivalent sample to the Republic of Ireland sample and 29 schools, 3% of all schools (9 primary and 20 post-primary), were recruited. Time constraints and examination period restrictions resulted in the need to convenience sample some schools (*n* = 5). Similar stratification profiles were used for convenience sampling. Once a school was recruited, every child within the specific year group was invited to participate. One year group was sampled from each of the 29 participating schools. Based on the total enrolments of the year groups sampled from each school, a participation rate of 60% was observed.

#### Sample Weighting

For the NI data, individual weights were calculated to avoid bias based on the proportion of pupils in the sample of a certain age and gender, referent to the relevant proportions in the NI school population. Information for weighting was obtained from the Department of Education (NI) [31].

### 2.3. Study Procedure

CSPPA 2018 was a multi-site study, undertaken by the University of Limerick, Dublin City University, University College Cork and Ulster University. The procedure for CSPPA was based on tried and tested methodology developed for schools [32]. Recruitment letters were distributed to the principals of all sampled schools, with researchers following up with a phone call within seven days. Once the school’s participation was confirmed, information and consent forms were distributed to all pupils who were eligible for participation in the study. Information about the study and “opt out” invitations were sent out two weeks in advance of the data collection. Pupils provided assent immediately prior to the survey starting and had the option to opt out at any point during the data collection process. 

Primary (including Year 6 and Year 7/ages 10–11 years old) and post-primary (Year 8–Year 14/age 12–18) pupils completed the survey between March and June 2018. The survey was conducted during school time and completed electronically via tablets, or in school computer suites, supervised by teachers and researchers. Before commencing the survey, the purpose of the study was briefly outlined, instructions were provided on how to complete the survey and applicable definitions were explained to all pupils. Participants were encouraged to reflect upon their answers, ask for assistance if required, and to be as honest as possible.

### 2.4. Outcome Measures

Survey questions were related to sport and PA across six domains: PA, PE, school sport, community sport, active travel and sedentary behaviour. Descriptions of questionnaire content are provided below and further information on sources and validation are included in the Appendix A. The variables associated with participation in these activities were also explored. Ethical approval was granted by the University of Limerick and recognised by all participating institutions (approval number 2017_11_19_EHS, date 16 January 2018). A process of passive consent was used, with pupils also providing their assent immediately prior to the survey starting. 

#### 2.4.1. Demographic Information

Socio-demographic information was collected at the individual level (i.e., age, gender, socio-economic status and disability status). Disability status was measured using the Child Functioning Module questionnaire [33,34] which assesses functional difficulties across domains including hearing, vision, communication, comprehension, learning, mobility and emotions on a rating scale (no difficulty, some difficulty, a lot of difficulty or cannot do at all). Participants were categorised as having no functional difficulty or having at least 1 functional difficulty.

Family socio-economic status was assessed using the Family Affluence Scale II (FAS II) [35], whereby respondents are classified as either low, medium or high FAS, with high FAS being those with the highest socio-economic status [35].

#### 2.4.2. Physical Activity

PA was assessed using the PACE+ two item screening tool [36]. Participants were asked to report how many days they had accumulated 60 min of MVPA in the last seven days, and over a typical 7-day period. The average of these was used to present habitual PA. The PACE+ tool has been shown to have acceptable validity for assessing adherence to the PA guidelines in adolescents [36,37,38].

#### 2.4.3. Enjoyment of Physical Activity

Enjoyment of PA was measured using the Physical Activity Enjoyment Scale (PACES) [39]. The PACES tool is a 16-item questionnaire that asks respondents to complete statements pertaining to why they may enjoy PA (e.g., “I get something out of it”), using a 5-point Likert scale ranging from 1 (disagree a lot) to 5 (agree a lot). 

#### 2.4.4. Physical Education, School Sport and Community Sport Participation

Questions designed to capture information on PE, school sport and community sport participation were consistent with the Economic and Social Research Institute (ESRI) survey [40] and the CSPPA 2010 survey [30].

Participants reported the frequency and duration of PE classes in curriculum time. Participants reported frequency of participation in school sport (school-based sport outside of curriculum time) and community sport (in non-school clubs) from the following options “4 or more days a week”, “2–3 days a week”, “once a week”, “2–3 days a month”, “one day a month”, “less often” and “never”. Participants were categorised as a “current participant” in school sport or community sport if they participated in either domain at least once per week. 

#### 2.4.5. Social Support

The social environment was assessed through frequency of peer and family support for involvement in PA. The scales were developed for the Amherst Health and Activity study [41] the 5 item family social support scale requires the participant to indicate how often a member of their family encouraged them to do PA, did PA with them, provided transportation to a place to do PA, watched them do PA or told them that that were doing well in PA or sport on a 6 point Likert scale ranging from never (0) to everyday (5). Scores on each item were then added for a minimum score of 0 and a maximum score of 25.

The peer measure asked the respondent to indicate frequency of encouraging friends to do PA, encouragement from friends to do PA, doing PA with friends, being teased by friends for not being good at PA (reversed scoring) and being told that you are doing well in PA or sport. The same Likert scale and scoring system was used.

### 2.5. Data Analysis

Data analysis was conducted using IBM SPSS Statistics (version 25; SPSS Inc., (version 25.0. Armonk, NY, USA)). Data were assessed for normality using the Kolmogorov–Smirnov statistic. Descriptive statistics (mean, standard deviation, frequency and percentage) were used for reasonably normally distributed data. Where data were not normally distributed, summary values are reported as median (Md) and interquartile range (IQR). Independent sample t-tests were used to compare differences between gender (male, female) and school (primary, post-primary) for average PA enjoyment scores.

Mann–Whitney U tests were used to compare differences between groups (male, female; primary, post-primary) for minutes of weekly PE. The Kruskal–Wallis test was used to compare differences in minutes of weekly PE across year groups at post-primary (more than 2 groups). Associations between categorical data were assessed using the Chi square test of independence. Direct logistic regression was used to assess the impact of a set of predictor variables on the likelihood of participants meeting the PA guidelines. The model contained five independent variables (age, gender, enjoyment of PA, social support (friends) and social support (parents)). Pupils were able to identify as male, female or other. Due to the small number of pupils who identified as other (2.2%), gender related outputs only included male and female. Probability was set at *p* < 0.05.

## 3. Results

### 3.1. Sample Size

In total, 1954 respondents completed the survey across nine primary schools (*n* = 446) and 20 post-primary schools (*n* = 1508).

### 3.2. Demographics

The majority of the schools surveyed (Table 1) were co-educational (81.4%). Less than a fifth of participants attended a single sex school (11.9% male and 6.6% female, respectively). The demographics of participants are shown in Table 2. Across the study sample in total, the majority of participants identified as Northern Irish (43%), Irish (31%) and British (20%). Within primary schools, just over half of respondents were male (51.1%), with a mean age of 10.57 years (SD = 0.51). In post-primary schools, 48.5% of respondents were male, with a mean sample age of 14.31 years (SD = 1.84). Chi square tests indicated no significant differences in the proportion of males and females across each age category.

Twenty-three percent of primary pupils were classified as low FAS, 54% as medium FAS and 23% as high FAS. Nineteen percent of post-primary were classified as low FAS, 58% as medium FAS and 23% as high FAS.

### 3.3. Physical Activity

Thirteen percent of participants met the PA guidelines of 60 min of MVPA per day. As shown in Figure 1, the proportion of participants achieving 60 min of MVPA daily was higher in primary school pupils (20%), compared with post-primary school pupils (11%). There appeared to be no association between gender and meeting PA guidelines among primary school pupils, with 21.1% of males meeting the guidelines as compared with 19.4% of females, x^2^ (1, *n* = 445) = 0.11, *p* = 0.743, phi = −0.021. There was a significant association between gender and meeting the guidelines at post-primary level, with males more likely to meet the guidelines when compared with their female counterparts (14.2% vs. 7.2%), x^2^ (1, *n* = 1466) = 18.14, *p* < 0.001, phi = −0.113. The likelihood of meeting the guidelines decreased with increasing age across the sample (*p* < 0.001). The proportion meeting the guidelines decreased with each increasing age category in females, however there was a slight increase between 10–11 years (20.8%) and 12–13 years (21.1%) in males.

There was no significant association between socio-economic status and meeting the guidelines among primary school pupils (low FAS = 20% vs. high FAS = 26%) and post-primary pupils (low FAS = 8% vs. high FAS = 14%).

### 3.4. Enjoyment and Support for Physical Activity

Enjoyment of PA was high among primary school participants, with an average score 68.9 ± 11.2 (range 16–80) (Table 3). Females reported higher PA enjoyment scores when compared with males (70.8 ± 9.8 vs. 67.1 ± 12.1, t(443) = −3.46, *p* < 0.001). Compared to primary school participants, PA enjoyment scores were lower amongst post-primary school participants, t(1909) = 18.30, *p* < 0.001), with a mean score of 58.7 ± 10.5 (range 20–76). There was no difference in reported enjoyment scores between post-primary males and females (59.0 ± 10.1 vs. 58.4 ± 11.0, *p* = 0.282). There were significant differences in social support from friends between males and females across both school levels, but no differences in family support between genders (Table 3).

A logistic regression analysis was used to assess the impact of predictor variables (gender, age, enjoyment of PA, social support (friends) and social support (parent)) on the odds that participants would meet the PA guidelines. The full model containing all predictors was statistically significant, x^2^ (5, *n* = 1911) = 217.20, *p* < 0.001. The model as a whole correctly classified 87.1% of cases. As shown in Table 4, all independent variables (excluding age) made a unique statistically significant contribution to the model. The strongest predictor for achieving the PA guidelines was gender (male). Older pupils were less likely to meet the PA guidelines (OR, 95% CI: 0.94, 0.88–1.01). Higher enjoyment of PA, and social support from family and friends were significant predictors of meeting the PA guidelines (Table 4).

### 3.5. Physical Education

The majority of primary school children (97.1%) reported having PE classes at least once per week, but only 7.2% of children reported achieving the recommended 120 min of PE per week. A Mann–Whitney U test indicated a significant difference in minutes of weekly PE at primary school level between Year 6 (Md = 50, *n* = 228) and Year 7 (Md = 50, *n* = 218) pupils, U = 0.5, z = 2.188, *p* = 0.029, r = 0.10. There were no significant differences between genders within each year group at primary school level. There was no difference across genders in the likelihood of meeting the recommended amount of PE. Schools with lower % FSM (Md = 60.0, IQR 45–90) reported significantly more minutes of PE than those with higher % FSM (Md = 30.0, IQR 30–41), *p* < 0.001.

Minutes of weekly PE per year group for post-primary are presented in Table 5. A Kruskal–Wallis test showed a statistically significant difference in minutes of weekly PE across all year groups (Year 8–14), x^2^ (6, *n* = 1508) = 449.8, *p* < 0.001. Post-hoc comparisons are outlined in Table 5. There were significant differences in minutes of PE between genders within Years 9, 10, 12 and 14 (Table 3).

At post-primary school level, 40.2% of participants achieved the PE guidelines; however, the proportion varied across year groups and gender (Figure 2). Males were also more likely than females to achieve the recommended 120 min per week of PE (45% vs. 34.3%), x^2^ (1, *n* = 1417) = 16.59, *p* < 0.001, phi = −0.110. Schools with lower %FSM were significantly more likely to meet PE recommendations than those with higher %FSM (40.2% vs. 20.8%, *p* < 0.001).

The content of males’ PE was dominated by football, with 69.4% of males reporting this sport during timetabled PE classes over the last 12 months. Other commonly cited sports/activities included athletics, baseball/rounders, basketball, Gaelic football/hurling and rugby. For females, the most common activity/sport during timetabled physical education was athletics. Other frequent activities included baseball/rounders, basketball, cross-country, dance and hockey.

### 3.6. Sport Participation

As outlined in Table 6, 64.5% of primary and 57.8% of post-primary participants reported taking part in school sport at least one day a week. At primary school level, being a current participant in school sport did not differ by gender, year group or school location. For post-primary school pupils, participation in school sport differed by gender (with males more likely to participate, *p* < 0.001) and year group (with younger pupils more likely to participate, *p* < 0.001). Post-primary pupils from a higher socio-economic status were more likely to participate in school sport than those from a lower socio-economic status (low FAS = 49% vs. high FAS = 64%, *p* < 0.001).

For community sport, 65.1% of primary and 48.6% of post-primary participants took part at least one day per week. At primary school level, reported current participation in community sport differed by socio-economic status, with those from high socio-economic status significantly more likely to participate in community sport than those from low socio-economic status (low FAS = 38% vs. high FAS = 70%, *p* < 0.001)

At post-primary school level, reported current participation in community sport differed by year group (e.g., Year 8 = 73% vs. Year 14 = 31%, *p* < 0.001), school location (rural = 51% vs. urban = 46%, *p* = 0.040) and socio-economic status (low FAS = 40% vs. high FAS = 55%, *p* = 0.004).

## 4. Discussion

The aim of this study was to investigate participation in PA across a range of contexts (overall PA, PE, school sport and community sport) in a representative sample of children and young people from NI.

This study reported low levels of adherence to PA guidelines (13% overall). The HBSC study examined PA levels amongst school children aged 11 and 15, from 45 countries [1]. Analysis of the findings from this study suggest that when compared to the HSBC data, NI children report lower levels of PA at both ages 11 (20 vs. 25%) and 15 (11% vs. 16%) years. While acknowledging that self-reported PA can vary depending upon the instrument used and comparisons, particularly between self–report survey methodologies, are problematic [42], the results from our investigation suggest that NI children are lagging behind their international peers by the time they leave primary school and the gap is sustained during adolescence. Our analysis is also consistent with the findings of other UK-wide research that reported on device measured PA and supports the contention that NI children are least likely in the UK to meet guidelines, at age 11 [9] in Europe [42,43] and internationally [44].

The results also support trends of age-related decline found in other cohort studies, which indicate a progressive decline in PA levels across primary school years [45,46]. A low proportion of NI primary school children were meeting the recommendations before 11 years and levels decline into adolescence, culminating in only 11% of post-primary school children meeting the PA guidelines.

A comparative analysis of gender differences in primary school pupils meeting daily MVPA guidelines (males 21.1% and females 19.4%) suggest only a small gender gap. Given that females are consistently reported in the literature as being less likely to meet the PA guidelines compared with their male counterparts [9,42,46], this finding is interesting. Furthermore, in comparison with their ROI peers, NI primary school females were more likely to meet PA guidelines (ROI 13% vs. NI 21%). Our analysis suggests that post-primary males are more likely to meet daily PA guidelines than females (10% and 16%) and the female decline in PA is significantly more dramatic than their male peers during adolescence.

Several authors have suggested that parents and friends may have a role in enhancing PA [23,24,25,26,47]. Similarly, this study found that social support from family and friends (assessed separately) were significant predictors for participants meeting the PA guidelines. These results emphasise the importance of both parental and peer support for PA as crucial considerations within interventions and approaches to the promotion of PA.

Our comparative analysis of enjoyment scores between primary males and females suggest that females report higher enjoyment levels than males. While enjoyment scores were lower in post-primary children, participants who reported higher levels of enjoyment were statistically more likely to meet the PA guidelines. Studies [27,48] reported that children who enjoy PA tended to be happier, healthier and more active. Conversely, a lack of enjoyment is cited as being a factor leading to dropout from sport and PA [49]. While enjoyment is subjective and individual, evidence-based models targeting a range of principles and practices that are known to enhance fun and enjoyment are emerging [49,50] and small to moderate evidence that school-based PA interventions can be effective in increasing PA enjoyment in children and adolescents has been reported [51]. Further research examining the associations between enjoyment and levels of daily MVPA, and factors affecting enjoyment of PA, is warranted. There is little doubt that this is an important variable for sustaining children’s participation in PA and should be a priority when designing interventions and programmes for youth.

While 97% of primary children reported having PE at least once per week, only 7% met the recommended weekly physical education guidelines of 120 min per week. In our analysis comparing adherence to PE guidelines, we found that post-primary children were 33% more likely to meet recommended guidelines than primary children (7% vs. 40% adherence). These results are consistent with the School Omnibus [52] and the Sport NI [19,53] baseline surveys. The extremely low levels of PE participation in post-16 children are a big concern. The magnitude of the decline in PE provision or participation in Years 13 and 14 found arguably reflects the absence of statutory PE requirements or guidelines for the post-16 curriculum in NI. The findings from this study relating to the age-related decline in PE participation are consistent with findings from English schools [54]. Consistent with our findings on PA levels, gender was not a factor in participation rates in primary school PE but at post-primary level males did receive more PE minutes per week and were 9% more likely to achieve the guidelines than females (45% vs. 34%).

Findings demonstrated that there is some breadth in the range of PE activities on offer in primary schools, the post-primary PE curriculum remains games dominated, particularly for males. It has been reported that the types of sports and activity a school provides appears to be a critical factor in understanding the differences in participation and the blend of activities on offer in extracurricular programmes appears to have an impact on female participation in particular [55]. Encouraging a broad and balanced curriculum, with a diverse range and number of activities in schools, is recommended [56].

Overall, it is evident that there is a need to address PE curriculum time allocation, adherence and provision, particularly in primary and post-16 age groups in NI schools. PE mandates have been shown to play an important role in promoting physical activity [57,58], a minimum statutory requirement of 120 min of PE may be beneficial in NI if the persistent low levels of curriculum time are to be addressed effectively.

Following analysis of data relating to sport participation in school and community settings, a healthy picture of participation emerged. The majority of children took part in primary (64.5%) and post-primary (57.8%) school sport at least once a week. Further comparative analysis suggests that while a similar trend of weekly community sport participation was seen in primary-aged children (65.1%), there was less participation reported in post-primary community sport settings (48.6%). Examination of the participation data by gender, in both school and community sport settings, suggests that the gender gap, where males were more likely than females to participate at least once per week, emerged in adolescence in both settings. Age-related decline for both males and females was evident and consistent with other cohort studies [22,44,59]. Overall, school and community sport participation are consistently lower for Northern Irish children than their Irish [29], UK [60] and international counterparts [61]. Given that children’s participation in sport is associated with higher levels of PA generally and lower levels of sedentary behaviour during the teenage years [62], it makes sense that promoting children’s participation in sports could be an effective public health strategy to help children meet the current PA guidelines. Further examination of the cultural or policy explanations for the lower rates of participation will facilitate greater understanding of how to encourage more NI children to take part in sport.

In this study, the overall picture emerging relative to gender is interesting. Research from Denmark found that girls were reported as being significantly less active in almost all domains and subdomains scrutinised [63]. Results from this study suggest that while female participation in PA, PE and sport (school and community) is lower overall, during the primary years, the gap is not as evident. While the primary school figures are encouraging in terms of gender balance, a rapid decline in female participation was evident in all contexts (PA, PE and sport) during adolescence. More research is needed to better understand the factors leading to the relatively strong levels of participation in primary-aged females and the subsequent and dramatic decline. The challenge is to develop an understanding of the associations and factors leading to this with a view to sustaining participation it into adolescence. Previous research has identified the strong macho culture of extra-curricular sport as a barrier to participation for females [64]. Females cite feelings of incompetence, a distaste for structured sport or lack of time as barriers to participation [56]. The need to focus on alternative sports or activities to competitive, team-based sports and the creation of opportunities to engage in single-sex activities may increase accessibility for females during adolescence [56].

An association between socio-economic status (SES) and PA among adolescents, where those with higher SES are more physically active than those with lower SES, is established in the literature [48]. An association between SES and PA among adolescents, where those with higher SES are more physically active than those with lower SES, is established in the literature [48]. It was evident in our analysis that in the overall picture, SES was not associated with meeting daily MVPA recommendations. A mixed picture of association between SES and participation emerged upon further analysis. SES does appear to be a determinant of participation levels in some settings. The data from this study suggest that pupils attending schools with low SES were more likely to receive the recommended 120 min of PE; however, beyond the school day children from lower SES groups were not participating in sport (school or community) to the same levels as their high SES peers. One of the aims of The NI Sport Matters Strategy 2009–2019 was to deliver a 6% increase in sport participation rates among socio-economically disadvantaged groups [65]. The issue of participation in sport being influenced by SES stubbornly persists further efforts to identify and remove barriers to participation need to be made.

Schools are ideally placed for the implementation of PA programmes as they have an extensive reach to youth populations and to children from all socio-economic backgrounds. The school environment can operate as an enabler and barrier to children’s PA [17,20,66]. A Cochrane Review of the WHO Health Promoting Schools Framework reported it to be effective in improving PA and physical fitness [67]. Schools are better able to enhance children’s PA when they emphasise resource provision for PA within the school day, create a ‘‘culture’’ of PA, train teachers to support a positive climate for PA promotion, and ensure extracurricular PA opportunities for all children [17,19]. When the school day is targeted and whole school approaches are adopted, opportunities for all can be positively impacted [66,67,68]. Health promoting school initiatives have operated in NI in the past and could provide a framework for change to help tackle some of the issues presented.

Social support, enjoyment, feelings of competence, the structure of sport and time have all been identified as factors that may impact participation [19,48,49]. Researchers advocate a nuanced approach to tackling the issues and given the complexity of the relationship between PE and youth sport, and lifelong participation in PA [69], an understanding of the significant variables are crucial to address some of the concerns with female participation and inequalities evidenced for children from lower SES backgrounds in particular.

Currently NI does not have a PA plan or strategy. It is evident that there is a need for such a plan to enable a more strategic and coordinated approach to challenge NI youth to find sustainable PA solutions in the context of increasingly sedentary lifestyles. Previous research reviewed national PA policies, identified characteristics for effective policy and implementation including multi-sectoral public health partnerships, and was informed by evidence, political support, long-term investment and commitment to programme implementation and evaluation [70]. Gathering effective surveillance of PA is an essential starting point to building a coordinated, targeted and systematic approach to tackling the complex issues of childhood physical inactivity. The current study provides a type of surveillance data and can contribute to the development of baseline figures for children’s sport and PA planning and development. The specific analysis of findings relating to associations between gender, SES, age, social support, enjoyment and PA may help in establishing appropriate targets for a range of sectors, and specific populations or domains that could underpin such a plan. Future and more frequent versions of the research could be used to monitor and track PA participation. In addition, the capacity of key sectors and stakeholders to implement a PA strategy will be central to success, further research that explores the implementation of PA in school and community settings is recommended.

## 5. Limitations

There are a number of limitations that should be taken into consideration when interpreting the findings of this study. Response bias of self-reported measurements may have occurred. Efforts to enhance the accuracy of responses by using developmentally appropriate and psychometrically valid self-report instruments, highly trained staff, supervision of respondents, and preservation of confidentiality and anonymity at all times, mitigated against this. Distinct from the Republic of Ireland part of this 2018 CSPPA study, physical health measures were not collected from the NI sample nor were any device-based PA measurements gathered. In future iterations of this study, objective measures of PA are recommended for use to further validate/substantiate the self-report findings.

The timing of the study coincided with the peak of the NI examination and assessment period, making access to participants challenging, resulted in the need to use convenience sampling. Efforts were made to ensure that similar stratification profiles were used for convenience-sampled schools and to mitigate any potential impact on the overall representativeness of the sample. The small number of primary schools sampled in this study (*n* = 9) may also impact the representativeness of this sample, despite all efforts being made to stratify as appropriate. The school sample in this study did not include any primary or post-primary specialist schools for children with disabilities. Future research should include children with mild, moderate and severe disabilities, so that associations between disability and PA can be examined in detail.

## 6. Conclusions

This study explored the current participation rates of Northern Irish children in PA, sport and physical education, providing cross-sectional data of participation and establishing a potential baseline for future PA surveillance in NI. PA levels in NI children are low, with most children failing to meet recommended guidelines for PA or physical education. Overall, in sports participation (both school and community-based), NI children lag behind their UK, Republic of Ireland and European peers with post-primary females being among the most physically inactive. An age-related decline in PA is evident across the domains, and will require innovative interventions to address individual, social and environmental barriers to physical activity and sport participation and to address the inequalities that exist.

## Figures and Tables

**Figure 1 ijerph-17-06849-f001:**
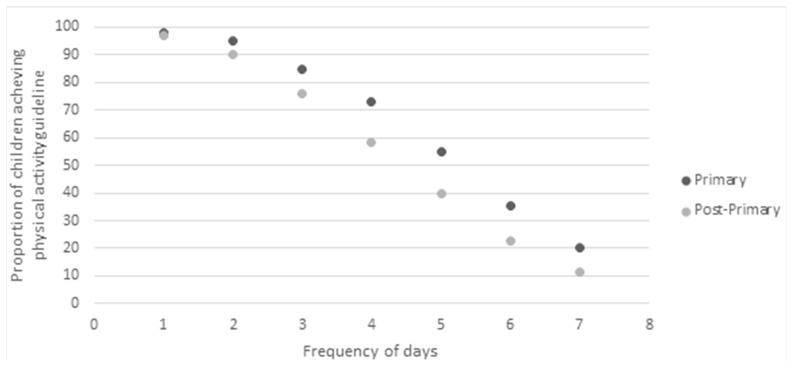
Proportion of primary and post-primary school children achieving 60 min of moderate-to-vigorous physical activity (MVPA) per day.

**Figure 2 ijerph-17-06849-f002:**
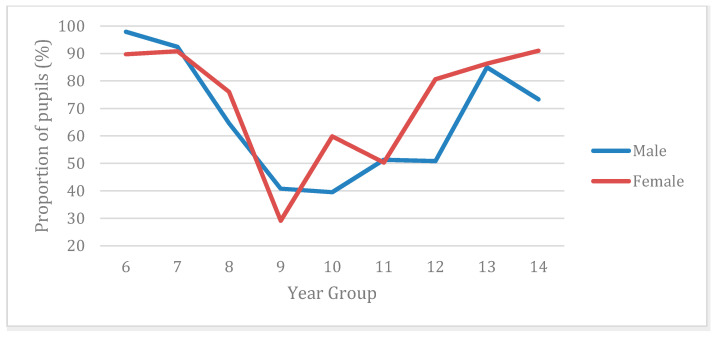
Proportion of males and females not achieving the recommended amount of physical education per year group.

**Table 1 ijerph-17-06849-t001:** Demographics of schools.

	Primary (*n* = 9)	Post-Primary (*n* = 20)
	*n* (%)	*n* (%)
**Gender**		
Mixed	9 (100)	16 (80)
Female	0 (0)	2 (10)
Male	0 (0)	2 (10)
**Location**		
Rural	4 (44)	9 (45)
Urban	5 (56)	11 (55)
**Size**		
Small	3 (33)	4 (20)
Medium	1 (11)	6 (30)
Large	5 (56)	10 (50)
**Socio-Economic Status ***		
Low	5 (56)	13 (65)
Medium	3 (33)	7 (35)
High	1 (11)	0 (0)
**Type of School**		
Controlled	8 (89)	9 (45)
Catholic Maintained	1 (11)	4 (20)
Voluntary	0 (0)	2 (10)
Integrated	0 (0)	5 (25)

* % Free Meals Class.

**Table 2 ijerph-17-06849-t002:** Demographics of participants.

	Primary	Post-Primary
	Total (*n* = 446) *n* (%)	Total (*n* = 1508) *n* (%)
**Gender**		
Male	228 (51.1)	731 (48.5)
Female	217 (48.7)	735 (48.7)
Other	1 (0.2)	42 (2.8)
**Age Category**		
10–11 years	442 (99.1)	38 (2.5)
12–13 years	4 (0.9)	497 (33.0)
14–15 years	NA	601 (39.9)
16–20 years	NA	372 (24.7)
**Disability Status**		
No functional difficulties	376 (84.3)	1216 (80.6)
At least 1 functional difficulty	70 (15.7)	292 (19.4)

**Table 3 ijerph-17-06849-t003:** Enjoyment and social support for physical activity.

	Overall	Male	Female
Enjoyment of PA			
Primary	68.9 ± 11.2	67.1 ± 12.1	70.8 ± 9.8 ^a^
Post-primary	58.7 ± 10.5 ^b^	59.0 ± 10.1	58.4 ± 11.0
Social support (friends)			
Primary	3.22 ± 0.7	3.13 ± 0.7	3.32 ± 0.7 ^c^
Post-primary	2.79 ± 0.6 ^b^	2.86 ± 0.7	2.73 ± 0.7 ^a^
Social support (family)			
Primary	3.20 ± 0.8	3.14 ± 0.8	3.26 ± 0.8
Post-primary	2.74 ± 1.0 ^b^	2.75 ± 1.1	2.72 ± 1.0

^a^ Significant difference between male and female, *p* < 0.001; ^b^ Significant difference between primary and post-primary, *p* < 0.001; ^c^ Significant difference between male and female, *p* < 0.05.

**Table 4 ijerph-17-06849-t004:** Logistic regression predicting likelihood of meeting the physical activity guidelines.

MEASURE	OR	95% CI	*p*
Gender (male)	1.80	1.35–2.41	<0.001
Age	0.94	0.88–1.01	0.100
Enjoyment of PA	1.03	1.02–1.05	<0.001
Social support (Friends)	1.11	1.05–1.16	<0.001
Social Support (Family)	1.12	1.07–1.16	<0.001

**Table 5 ijerph-17-06849-t005:** Minutes of weekly physical education per year group (post-primary).

	Male		Female		Total Sample	
	Median	IQR *	Median	IQR	Median	IQR
Year 8	90	60–120	80	55–105	90 ^a,b,c,d,e,f^	60–120
Year 9	140 ^g^	90–200	160	105–280	160 ^b,c,d,e,f,h^	105–240
Year 10	140 ^g^	80–160	80	50–160	120 ^a,e,f,h^	70–160
Year 11	105	90–180	108	90–150	105 ^a,e,f,h^	90–160
Year 12	105 ^g^	100–200	95	90–105	105 ^a,e,f,h^	90–150
Year 13	0	0	0	0	0 ^a,b,c,d,h^	0
Year 14	0 ^g^	0	0	0	0 ^a,b,c,d,h^	0

* Interquartile Range (IQR) ^a^ Significantly different to Year 9, (*p* < 0.001); ^b^ Significantly different to Year 10, (*p* < 0.001); ^c^ Significantly different to Year 11, (*p* < 0.001); ^d^ Significantly different to Year 12, (*p* < 0.001); ^e^ Significantly different to Year 13, (*p* < 0.001); ^f^ Significantly different to Year 14, (*p* < 0.001); ^g^ Significant difference within year group for gender (*p* < 0.05); ^h^ Significantly different to Year 8, (*p* < 0.001).

**Table 6 ijerph-17-06849-t006:** Frequency of school sport and community sport participation for primary and post-primary school pupils.

	Primary Total (*n* = 446)		Post-Primary Total (*n* = 1508)	
	*n*%		*n*%	
**School Sport**				
4 or more days a week	115	25.7	248	16.4
2–3 days a week	113	25.3	412	27.3
One day a week	60	13.5	213	14.1
Less often	55	12.3	92	6.1
Never	103	23.2	543	36.1
**Community Sport**				
4 or more days a week	58	12.9	273	18.1
2–3 days a week	138	30.9	334	22.2
One day a week	95	21.3	126	8.3
Less often	92	20.7	72	4.8
Never/not applicable	63	14.2	703	46.6

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
