# Peer review of "Physical Activity, Sport and Physical Education in Northern Ireland School Children: A Cross-Sectional Study"

_ijerph, 2020, doi:10.3390/ijerph17186849_

Round 1
Reviewer 1 Report
Dear Authors
the revised version has been improved the paper.
In my point of view only any abbreviation must be define before to use also in the text
Author Response
REVIEWER 1
POINT 1
In my point of view only any abbreviation must be define before to use also in the text
RESPONSE POINT 1
Thank you for your comments. We have reviewed the entire manuscript and checked all abbreviations and made any corrections as appropriate.
Reviewer 2 Report
See attach

Reviewer 3 Report
This cross-sectional study provides data on self-reported physical activity, sport and physical education participation of school children in Northern Ireland. The authors concluded that females and children from lower socio-economic backgrounds are less likely to achieve the physical activity guidelines, suggesting the need for specific interventions. Although the topic is generally worthy of consideration, the poor representation of the results militates against the publication of the manuscript in the current form. Detailed comments are listed below.
The results of the statistical analyzes are reported exclusively in the text, while the tables only show descriptive statistics; this compromises the readability and interpretation of the results by readers. Therefore, the main information presented in the text should be included in new tables and scrupulously accompanied by all the necessary statistical data (e.g. t, X2, r).
Again, the manuscript largely lacks illustrative representation by summarizing correlations in the text without showing the original data but rather an r/p value. This is a poor representation of scientific data. Some graphs with correlations are highly recommended and should be included.
Lastly, the authors declare that they only used SPSS, when in reality the two figures were made with Microsoft Excel. In addition to removing Figure 1, replacing it with a scatterplot, Figure 2 should be made with SPSS; in this regard, SPSS Version 25 should be cited as IBM SPSS Statistics (Version 25.0. Armonk, NY).
Overall, the study has important application and may have the potential to add meaningful information to the current body of literature.
Author Response
Please see the attachment
Much obliged
Sinead Connolly

Round 2
Reviewer 2 Report
R2:
The authors made the required changes.
This manuscript is a resubmission of an earlier submission. The following is a list of the peer review reports and author responses from that submission.
Round 1
Reviewer 1 Report
Dear authors
the paper entitled "Physical activity, sport and physical education in Northern Ireland school children: A cross-sectional study" is an intersesting manuscript that describe physical activity level in NI children. The strenght of the paper is the description of PA across general PA, PE, school sport and community sport with an additional focus on enjoyment and socio-economic status.
In my opinion some minor aspects should be addressed:
- The introduction appears too general, there is a lot of information not all relevant to provide the background of the study which does not focus the reader on the purpose of the study. The reviewer suggests reducing this section to a more concise version.
- the reviewer suggest to choose keywords different from title.
- at line 53 please define "NI"
- at line 54 please choose between MVPA or full define
- please check at line 104 "or numerals in square brackets, e.g., [1] or [2,3], or [4–6]. See the end of the document for further details on references"
- please check at line 287-288, 296-297 the abbreviation FMS (FAS?) not previously define
- Why in table 3 year 13-14 haven't any results? In figure 2 some data are shown for PE. Maybe the median does not seem the appropriate way to present these results.
- Please check the results in table 4. In some cases the total is not 100% or the sum of the cases is not the sample number shown. in addition, it is advisable to indicate an additional section in which the results relating to school sport and community sport are jointly described for a global vision of the individual subjects.
- please check at line 409 "SS"
Reviewer 2 Report
See attach

Reviewer 3 Report
Connolly and co-workers considered an important topic since insufficient physical activity and sedentary time are a major health problem globally. I have some minor revisions to suggest in order to improve the manuscript quality:
Introduction
Page 3 line 104. The sentence is not clear I think a part is missing.
Material and Methods
Page 3 line 118. Since it is unusual, please describe the motivation for choosing the percentage of free school meals as criteria for determining socioeconomic status.
Page 4 line 164 I doubt that it is correct to consider in the same group children having different functional difficulties
Page 5 line 208. it is not well specified which variables have been included in the t-test. If there are more than 2, as I think, an ANOVA test is recommended where the p-value must be adjusted according to a post hoc test. In addition, more information should be added in the results section, especially for subgroups comparisons.
Methods
As stated by the authors (l.108-113), the main results of the “The Children’s Sport Participation and Physical Activity Study 2018” are already published and available online (https://www.sportireland.ie/sites/default/files/2019-10/csppa-2018-final-report_1.pdf) and it makes me doubt about the contribution that the publication can make to the existing literature.
Results
Table 2. Is sex distribution homogeneous in all the age classes?
Page 7 line 247. “The likelihood of meeting the guidelines decreased within increasing age across the sample (p<0.001)”. Is this true for both sexes and with the same trend?
Page 7 line 249. What kind of analysis was carried out? Since there are three categories, I think that a pairwise comparison is not correct.
Page 8 line 245. Is correct to consider in the same kind of activities both for children with disabilities or not? Did children with disabilities follow specific activities? It sounds difficult to have the same criteria for comparison.
Discussion
The discussion section is very descriptive and offers limited comparisons to previous research. Therefore, in order to implement this section I suggest considering the following papers:
Participation in sports, body composition, and fitness characteristics in children according to ethnic background.
Toselli S, Belcastro MG.Scand J Med Sci Sports. 2017 Dec;27(12):1913-1926. doi: 10.1111/sms.12843. Epub 2017 Feb 24.PMID: 28107555
Do extra compulsory physical education lessons mean more physically active children--findings from the childhood health, activity, and motor performance school study Denmark (The CHAMPS-study DK).
Møller NC, Tarp J, Kamelarczyk EF, Brønd JC, Klakk H, Wedderkopp N.Int J Behav Nutr Phys Act. 2014 Sep 24;11:121. doi: 10.1186/s12966-014-0121-0.
Changes in physical activity and sedentary time during adolescence: Gender differences during weekdays and weekend days.
Kallio J, Hakonen H, Syväoja H, Kulmala J, Kankaanpää A, Ekelund U, Tammelin T.Scand J Med Sci Sports. 2020 Jul;30(7):1265-1275. doi: 10.1111/sms.13668. Epub 2020 Apr 20.
References
Finally, the authors did not bother to observe the guidelines required by IJERPH. This should be done carefully when submitting an article to a Journal that requests and accepts only high-quality papers.
Reviewer 4 Report
Please ensure all abbreviations are capitalised first use e.g. Physical Activity (PA), check whole document.
Insert comma in abstract: “Findings suggest that PA levels are low, with a minority of children (13%) meeting the PA guidelines (primary pupils 20%, post primary pupils 11%).
The introduction is clear and well written with relevance evidence cited to justify current study.
Check brackets: Line 122 “ total of 51 schools were contacted (20 primary and 31 post-primary to ensure an equivalent sample to the Republic of Ireland sample and 29 schools (9 primary and 20 post-primary, 3% of all schools) were recruited.
Of the 29 schools recruited into the study, what was the potential sample size of students, so to later calculate the participation rate e.g. 1954/x?
Given the acknowledgment of the smaller than desired sample size included due to convenience sampling, it would be helpful within the results to include the study power and effect sizes to help demonstrate representation
Whilst acknowledging the need for a NI Strategy and that schools are key environments to promote PE/PA, it would be noteworthy to reflect, the ability of schools to provide this: in terms of skill, time, resources. (Whilst a strategy in itself would be helpful to direct, the schools technically could do this if it was considered a priority topic- is there are need for further research to explore the implementation of PE/PA within school curriculum/ environment.)
Overall a well written and clear paper.